# FEDERATED LEARNING: STRATEGIES FOR IMPROVING COMMUNICATION EFFICIENCY

## ABSTRACT

Federated Learning is a machine learning setting where the goal is to train a high-quality centralized model while training data remains distributed over a large number of clients each with unreliable and relatively slow network connections. We consider learning algorithms for this setting where on each round, each client independently computes an update to the current model based on its local data, and communicates this update to a central server, where the client-side updates are aggregated to compute a new global model. The typical clients in this setting are mobile phones, and communication efficiency is of the utmost importance.

In this paper, we propose two ways to reduce the uplink communication costs: *structured updates*, where we directly learn an update from a restricted space parametrized using a smaller number of variables, e.g. either low-rank or a random mask; and *sketched updates*, where we learn a full model update and then compress it using a combination of quantization, random rotations, and subsampling before sending it to the server. Experiments on both convolutional and recurrent networks show that the proposed methods can reduce the communication cost by two orders of magnitude.

## 1 INTRODUCTION

As datasets grow larger and models more complex, training machine learning models increasingly requires distributing the optimization of model parameters over multiple machines. Existing machine learning algorithms are designed for highly controlled environments (such as data centers) where the data is distributed among machines in a balanced and i.i.d. fashion, and high-throughput networks are available.

Recently, Federated Learning (and related decentralized approaches) (McMahan & Ramage, 2017; Konečný et al., 2016; McMahan et al., 2017; Shokri & Shmatikov, 2015) have been proposed as an alternative setting: a shared global model is trained under the coordination of a central server, from a federation of participating devices. The participating devices (clients) are typically large in number and have slow or unstable internet connections. A principal motivating example for Federated Learning arises when the training data comes from users' interaction with mobile applications. Federated Learning enables mobile phones to collaboratively learn a shared prediction model while keeping all the training data on device, decoupling the ability to do machine learning from the need to store the data in the cloud. The training data is kept locally on users' mobile devices, and the devices are used as nodes performing computation on their local data in order to update a global model. This goes beyond the use of local models that make predictions on mobile devices, by bringing model training to the device as well. The above framework differs from conventional distributed machine learning (Reddi et al., 2016; Ma et al., 2017; Shamir et al., 2014; Zhang & Lin, 2015; Dean et al., 2012; Chilimbi et al., 2014) due to the very large number of clients, highly unbalanced and non-i.i.d. data available on each client, and relatively poor network connections. In this work, our focus is on the last constraint, since these unreliable and asymmetric connections pose a particular challenge to practical Federated Learning.

For simplicity, we consider synchronized algorithms for Federated Learning where a typical round consists of the following steps:

1. A subset of existing clients is selected, each of which downloads the current model.

2. Each client in the subset computes an updated model based on their local data.
3. The model updates are sent from the selected clients to the sever.
4. The server aggregates these models (typically by averaging) to construct an improved global model.

A naive implementation of the above framework requires that each client sends a full model (or a full model update) back to the server in each round. For large models, this step is likely to be the bottleneck of Federated Learning due to multiple factors. One factor is the asymmetric property of internet connection speeds: the uplink is typically much slower than downlink. The US average broadband speed was 55.0Mbps download vs. 18.9Mbps upload, with some internet service providers being significantly more asymmetric, e.g., Xfinity at 125Mbps down vs. 15Mbps up (speedtest.net, 2016). Additionally, existing model compressions schemes such as Han et al. (2015) can reduce the bandwidth necessary to download the current model, and cryptographic protocols put in place to ensure no individual client's update can be inspected before averaging with hundreds or thousands of other updates (Bonawitz et al., 2017) further increase the amount of bits that need to be uploaded.

It is therefore important to investigate methods which can reduce the uplink communication cost. In this paper, we study two general approaches:

- *Structured updates*, where we directly learn an update from a restricted space that can be parametrized using a smaller number of variables.
- *Sketched updates*, where we learn a full model update, then compress it before sending to the server.

These approaches, explained in detail in Sections 2 and 3, can be combined, e.g., first learning a structured update and sketching it; we do not experiment with this combination in this work though.

In the following, we formally describe the problem. The goal of Federated Learning is to learn a model with parameters embodied in a real matrix[1] $\mathbf{W} \in \mathbb{R}^{d_1 \times d_2}$ from data stored across a large number of clients. We first provide a communication-naive version of the Federated Learning. In round $t \geq 0$, the server distributes the current model $\mathbf{W}_t$ to a subset $S_t$ of $n_t$ clients. These clients independently update the model based on their local data. Let the updated local models be $\mathbf{W}_t^1, \mathbf{W}_t^2, \ldots, \mathbf{W}_t^{n_t}$, so the update of client $i$ can be written as $\mathbf{H}_t^i := \mathbf{W}_t^i - \mathbf{W}_t$, for $i \in S_t$. These updates could be a single gradient computed on the client, but typically will be the result of a more complex calculation, for example, multiple steps of stochastic gradient descent (SGD) taken on the client's local dataset. In any case, each selected client then sends the update back to the sever, where the global update is computed by aggregating[2] all the client-side updates:

$$\mathbf{W}_{t+1} = \mathbf{W}_t + \eta_t \mathbf{H}_t, \qquad \mathbf{H}_t := \tfrac{1}{n_t} \sum_{i \in S_t} \mathbf{H}_t^i.$$

The sever chooses the learning rate $\eta_t$. For simplicity, we choose $\eta_t = 1$.

In Section 4, we describe Federated Learning for neural networks, where we use a separate 2D matrix $\mathbf{W}$ to represent the parameters of each layer. We suppose that $\mathbf{W}$ gets right-multiplied, i.e., $d_1$ and $d_2$ represent the output and input dimensions respectively. Note that the parameters of a fully connected layer are naturally represented as 2D matrices. However, the kernel of a convolutional layer is a 4D tensor of the shape #input × width × height × #output. In such a case, $\mathbf{W}$ is reshaped from the kernel to the shape (#input × width × height) × #output.

**Outline and summary.** The goal of increasing communication efficiency of Federated Learning is to reduce the cost of sending $\mathbf{H}_t^i$ to the server, while learning from data stored across large number of devices with limited internet connection and availability for computation. We propose two general classes of approaches, structured updates and sketched updates. In the Experiments section, we evaluate the effect of these methods in training deep neural networks.

In simulated experiments on CIFAR data, we investigate the effect of these techniques on the convergence of the Federated Averaging algorithm (McMahan et al., 2017). With only a slight degradation in convergence speed, we are able to reduce the total amount of data communicated by two orders of magnitude. This lets us obtain a good prediction accuracy with an all-convolutional model, while in total communicating less information than the size of the original CIFAR data. In a larger realistic

---

[1] For sake of simplicity, we discuss only the case of a single matrix since everything carries over to setting with multiple matrices, for instance corresponding to individual layers in a deep neural network.

[2] A weighted sum might be used to replace the average based on specific implementations.

experiment on user-partitioned text data, we show that we are able to efficiently train a recurrent neural network for next word prediction, before even using the data of every user once. Finally, we note that we achieve the best results including the preprocessing of updates with structured random rotations. Practical utility of this step is unique to our setting, as the cost of applying the random rotations would be dominant in typical parallel implementations of SGD, but is negligible, compared to the local training in Federated Learning.

## 2 STRUCTURED UPDATE

The first type of communication efficient update restricts the updates $\mathbf{H}_t^i$ to have a *pre-specified structure*. Two types of structures are considered in the paper: *low rank* and *random mask*. It is important to stress that we train directly the updates of this structure, as opposed to approximating/sketching general updates with an object of a specific structure — which is discussed in Section 3.

**Low rank.** We enforce every update to local model $\mathbf{H}_t^i \in \mathbb{R}^{d_1 \times d_2}$ to be a low rank matrix of rank at most $k$, where $k$ is a fixed number. In order to do so, we express $\mathbf{H}_t^i$ as the product of two matrices: $\mathbf{H}_t^i = \mathbf{A}_t^i \mathbf{B}_t^i$, where $\mathbf{A}_t^i \in \mathbb{R}^{d_1 \times k}$, $\mathbf{B}_t^i \in \mathbb{R}^{k \times d_2}$. In subsequent computation, we generated $\mathbf{A}_t^i$ randomly and consider a constant during a local training procedure, and we optimize only $\mathbf{B}_t^i$. Note that in practical implementation, $\mathbf{A}_t^i$ can in this case be compressed in the form of a random seed and the clients only need to send trained $\mathbf{B}_t^i$ to the server. Such approach immediately saves a factor of $d_1/k$ in communication. We generate the matrix $\mathbf{A}_t^i$ afresh in each round and for each client independently.

We also tried fixing $\mathbf{B}_t^i$ and training $\mathbf{A}_t^i$, as well as training both $\mathbf{A}_t^i$ and $\mathbf{B}_t^i$; neither performed as well. Our approach seems to perform as well as the best techniques considered in Denil et al. (2013), without the need of any hand-crafted features. An intuitive explanation for this observation is the following. We can interpret $\mathbf{B}_t^i$ as a projection matrix, and $\mathbf{A}_t^i$ as a reconstruction matrix. Fixing $\mathbf{A}_t^i$ and optimizing for $\mathbf{B}_t^i$ is akin to asking "Given a given random reconstruction, what is the projection that will recover most information?". In this case, if the reconstruction is full-rank, the projection that recovers space spanned by top $k$ eigenvectors exists. However, if we randomly fix the projection and search for a reconstruction, we can be unlucky and the important subspaces might have been projected out, meaning that there is no reconstruction that will do as well as possible, or will be very hard to find.

**Random mask.** We restrict the update $\mathbf{H}_t^i$ to be a sparse matrix, following a pre-defined random sparsity pattern (i.e., a random mask). The pattern is generated afresh in each round and for each client independently. Similar to the low-rank approach, the sparse pattern can be fully specified by a random seed, and therefore it is only required to send the values of the non-zeros entries of $\mathbf{H}_t^i$, along with the seed.

## 3 SKETCHED UPDATE

The second type of updates addressing communication cost, which we call *sketched*, first computes the full $\mathbf{H}_t^i$ during local training without any constraints, and then approximates, or encodes, the update in a (lossy) compressed form before sending to the server. The server decodes the updates before doing the aggregation. Such sketching methods have application in many domains (Woodruff, 2014). We experiment with multiple tools in order to perform the sketching, which are mutually compatible and can be used jointly:

**Subsampling.** Instead of sending $\mathbf{H}_t^i$, each client only communicates matrix $\hat{\mathbf{H}}_t^i$ which is formed from a random subset of the (scaled) values of $\mathbf{H}_t^i$. The server then averages the subsampled updates, producing the global update $\hat{\mathbf{H}}_t$. This can be done so that the average of the sampled updates is an unbiased estimator of the true average: $\mathbb{E}[\hat{\mathbf{H}}_t] = \mathbf{H}_t$. Similar to the random mask structured update, the mask is randomized independently for each client in each round, and the mask itself can be stored as a synchronized seed.

**Probabilistic quantization.** Another way of compressing the updates is by *quantizing* the weights.

We first describe the algorithm of quantizing each scalar to one bit. Consider the update $\mathbf{H}_t^i$, let $h = (h_1, \ldots, h_{d_1 \times d_2}) = \text{vec}(\mathbf{H}_t^i)$, and let $h_{\max} = \max_j(h_j)$, $h_{\min} = \min_j(h_j)$. The compressed update of $h$, denoted by $\tilde{h}$, is generated as follows:

$$\tilde{h}_j = \begin{cases} h_{\max}, & \text{with probability} \quad \frac{h_j - h_{\min}}{h_{\max} - h_{\min}} \\ h_{\min}, & \text{with probability} \quad \frac{h_{\max} - h_j}{h_{\max} - h_{\min}} \end{cases}.$$

It is easy to show that $\tilde{h}$ is an unbiased estimator of $h$. This method provides $32\times$ of compression compared to a 4 byte float. The error incurred with this compression scheme was analysed for instance in Suresh et al. (2017), and is a special case of protocol proposed in Konečný & Richtárik (2016).

One can also generalize the above to more than 1 bit for each scalar. For $b$-bit quantization, we first equally divide $[h_{\min}, h_{\max}]$ into $2^b$ intervals. Suppose $h_i$ falls in the interval bounded by $h'$ and $h''$. The quantization operates by replacing $h_{\min}$ and $h_{\max}$ of the above equation by $h'$ and $h''$, respectively. Parameter $b$ then allows for simple way of balancing accuracy and communication costs.

Another quantization approach also motivated by reduction of communication while averaging vectors was recently proposed in Alistarh et al. (2016). Incremental, randomized and distributed optimization algorithms can be similarly analysed in a quantized updates setting (Rabbat & Nowak, 2005; Golovin et al., 2013; Gamal & Lai, 2016).

**Improving the quantization by structured random rotations.** The above 1-bit and multi-bit quantization approach work best when the scales are approximately equal across different dimensions.

For example, when $\max = 1$ and $\min = -1$ and most of values are 0, the 1-bit quantization will lead to a large error. We note that applying a random rotation on $h$ before the quantization (multiplying $h$ by a random orthogonal matrix) solves this issue. This claim has been theoretically supported in Suresh et al. (2017). In that work, is shows that the structured random rotation can reduce the quantization error by a factor of $\mathcal{O}(d/\log d)$, where $d$ is the dimension of $h$. We will show its practical utility in the next section.

In the decoding phase, the server needs to perform the inverse rotation before aggregating all the updates. Note that in practice, the dimension of $h$ can easily be as high as $d = 10^6$ or more, and it is computationally prohibitive to generate ($\mathcal{O}(d^3)$) and apply ($\mathcal{O}(d^2)$) a general rotation matrix. Same as Suresh et al. (2017), we use a type of structured rotation matrix which is the product of a Walsh-Hadamard matrix and a binary diagonal matrix. This reduces the computational complexity of generating and applying the matrix to $\mathcal{O}(d)$ and $\mathcal{O}(d\log d)$, which is negligible compared to the local training within Federated Learning.

## 4 EXPERIMENTS

We conducted experiments using Federated Learning to train deep neural networks for two different tasks. First, we experiment with the CIFAR-10 image classification task (Krizhevsky, 2009) with convolutional networks and artificially partitioned dataset, and explore properties of our proposed algorithms in detail. Second, we use more realistic scenario for Federated Learning — the public Reddit post data (Google BigQuery), to train a recurrent network for next word prediction.

The Reddit dataset is particularly useful for simulated Federated Learning experiments, as it comes with natural per-user data partition (by author of the posts). This includes many of the characteristics expected to arise in practical implementation. For example, many users having relatively few data points, and words used by most users are clustered around a specific topic of interest of the particular user.

In all of our experiments, we employ the Federated Averaging algorithm (McMahan et al., 2017), which significantly decreases the number of rounds of communication required to train a good model. Nevertheless, we expect our techniques will show a similar reduction in communication costs when applied to a synchronous distributed SGD, see for instance Alistarh et al. (2016). For Federated Averaging, on each round we select multiple clients uniformly at random, each of which performs several epochs of SGD with a learning rate of $\eta$ on their local dataset. For the structured

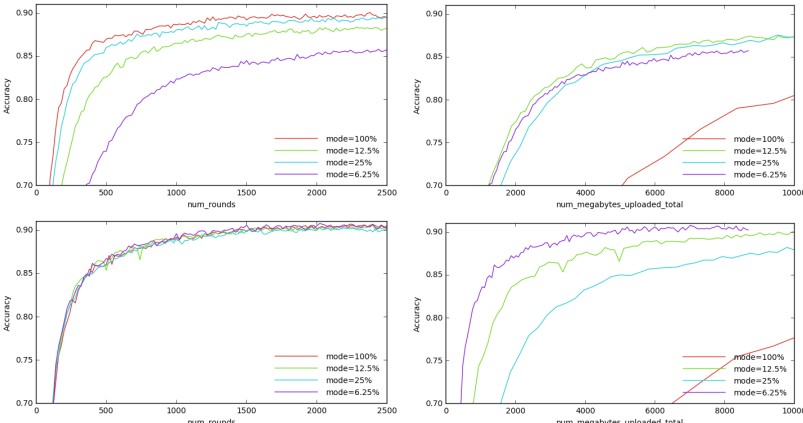

Figure 1: Structured updates with the CIFAR data for size reduction various modes. *Low rank* updates in top row, *random mask* updates in bottom row.

updates, SGD is restricted to only update in the restricted space, that is, only the entries of $\mathbf{B}_t^i$ for low-rank updates and the unmasked entries for the random-mask technique. From this updated model we compute the updates for each layer $\mathbf{H}_t^i$. In all cases, we run the experiments with a range of choices of learning rate, and report the best result.

## 4.1 CONVOLUTIONAL MODELS ON THE CIFAR-10 DATASET

In this section we use the CIFAR-10 dataset to investigate the properties of our proposed methods as part of Federated Averaging algorithm.

There are $50\,000$ training examples in the CIFAR-10 dataset, which we randomly partitioned into 100 clients each containing 500 training examples. The model architecture we used was the all-convolutional model taken from what is described as "Model C" in Springenberg et al. (2014), for a total of over $10^6$ parameters. While this model is not the state-of-the-art, it is sufficient for our needs, as our goal is to evaluate our compression methods, not to achieve the best possible accuracy on this task.

The model has 9 convolutional layers, first and last of which have significantly fewer parameters than the others. Hence, in this whole section, when we try to reduce the size the individual updates, we only compress the inner 7 layers, each of which with the same parameter[3]. We denote this by keyword 'mode', for all approaches. For *low rank* updates, 'mode = $25\%$' refers to the rank of the update being set to $1/4$ of rank of the full layer transformation, for *random mask* or *sketching*, this refers to all but $25\%$ of the parameters being zeroed out.

In the first experiment, summarized in Figure 1, we compare the two types of structured updates introduced in Section 2 — *low rank* in the top row and *random mask* in the bottom row. The main message is that *random mask* performs significantly better than *low rank*, as we reduce the size of the updates. In particular, the convergence speed of *random mask* seems to be essentially unaffected when measured in terms of number of rounds. Consequently, if the goal was to only minimize the upload size, the version with reduced update size is a clear winner, as seen in the right column.

In Figure 2, we compare the performance of structured and sketched updates, without any quantization. Since in the above, the structured *random mask* updates performed better, we omit *low rank* update for clarity from this comparison. We compare this with the performance of the sketched updates, with and without preprocessing the update using random rotation, as described in Section 3, and for two different modes. We denote the randomized Hadamard rotation by 'HD', and no rotation by 'I'.

---

[3]We also tried reducing the size of all 9 layers, but this yields negligible savings in communication, while it slightly degraded convergence speed.

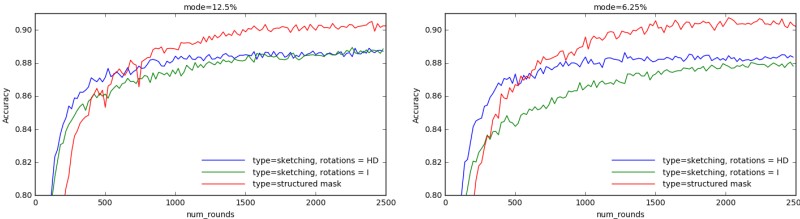

Figure 2: Comparison of structured *random mask* updates and sketched updates without quantization on the CIFAR data.

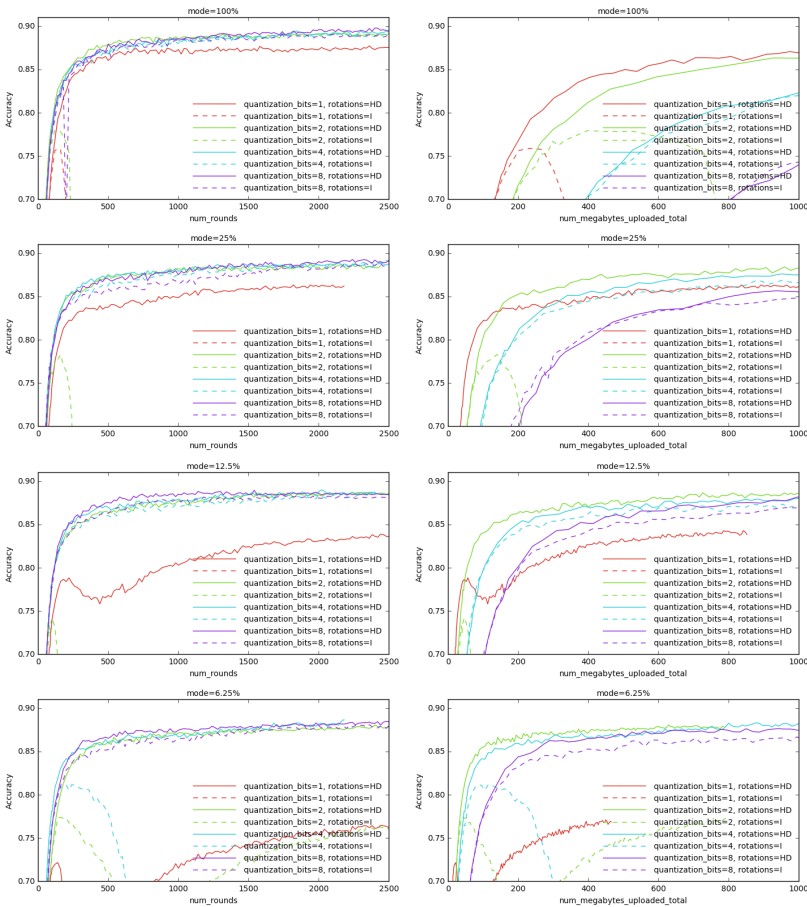

Figure 3: Comparison of sketched updates, combining preprocessing the updates with rotations, quantization and subsampling on the CIFAR data.

The intuitive expectation is that directly learning the structured *random mask* updates should be better than learning an unstructured update, which is then sketched to be represented with the same number of parameters. This is because by sketching we throw away some of the information obtained during training. The fact that with sketching the updates, we should converge to a slightly lower accuracy can be theoretically supported, using analogous argument as carefully stated in (Alistarh et al., 2016), since sketching the updates increases the variance directly appearing in convergence analysis. We see this behaviour when using the structured *random mask* updates, we are able to eventually converge to slightly higher accuracy. However, we also see that with sketching the updates, we are able to attain modest accuracy (e.g. 85%) slightly faster.

In the last experiment on CIFAR data, we focus on interplay of all three elements introduced in Section 3 — subsampling, quantization and random rotations. Note that combination of all these tools will enable higher compression rate than in the above experiments. Each pair of plots in Figure 3 focuses on particular mode (subsampling), and in each of them we plot performance with different bits used in quantization, with or without the random rotations. What we can see consistently in all plots, is that the random rotation improves the performance. In general, the behaviour of the algorithm is less stable without the rotations, particularly with small number of quantization bits and smaller modes.

In order to highlight the potential of communication savings, note that by preprocessing with the random rotation, sketching out all but $6.25\%$ elements of the update and using 2 bits for quantization, we get only a minor drop in convergence, while saving factor of 256 in terms of bits needed to represent the updates to individual layers. Finally, if we were interested in minimizing the amount of data uploaded, we can obtain a modest accuracy, say $85\%$, while in total communicating less than half of what would be required to upload the original data.

## 4.2 LSTM Next-Word Prediction on Reddit Data

We constructed the dataset for simulating Federated Learning based on the data containing publicly available posts/comments on Reddit (Google BigQuery), as described by Al-Rfou et al. (2016). Critically for our purposes, each post in the database is keyed by an author, so we can group the data by these keys, making the assumption of one client device per author. Some authors have a very large number of posts, but in each round of FedAvg we process at most $32\,000$ tokens per user. We omit authors with fewer than $1600$ tokens, since there is constant overhead per client in the simulation, and users with little data don't contribute much to training. This leaves a dataset of $763\,430$ users, with an average of $24\,791$ tokens per user. For evaluation, we use a relatively small test set of $75\,122$ tokens formed from random held-out posts.

Based on this data, we train a LSTM next word prediction model. The model is trained to predict the next word given the current word and a state vector passed from the previous time step. The model works as follows: word $s_t$ is mapped to an embedding vector $e_t \in \mathbb{R}^{96}$, by looking up the word in a dictionary of $10\,017$ words (tokens). $e_t$ is then composed with the state emitted by the model in the previous time step $s_{t1} \in \mathbb{R}^{256}$ to emit a new state vector $s_t$ and an "output embedding" $o_t \in \mathbf{R}^{96}$. The output embedding is scored against the embedding of each item in the vocabulary via inner product, before being normalized via softmax to compute a probability distribution over the vocabulary. Like other standard language models, we treat every input sequence as beginning with an implicit "BOS" (beginning of sequence) token and ending with an implicit "EOS" (end of sequence) token. Unlike standard LSTM language models, our model uses the same learned embedding for both the embedding and softmax layers. This reduces the size of the model by about 40% for a small decrease in model quality, an advantageous tradeoff for mobile applications. Another change from many standard LSTM RNN approaches is that we train these models to restrict the word embeddings to have a fixed L2 norm of 1.0, a modification found to improve convergence time. In total the model has 1.35M parameters.

In order to reduce the size of the update, we sketch all the model variables except some small variables (such as biases) which consume less than 0.01% of memory. We evaluate using `AccuracyTop1`, the probability that the word to which the model assigns highest probability is correct. We always count it as a mistake if the true next word is not in the dictionary, even if the model predicts 'unknown'.

In Figure 4, we run the Federated Averaging algorithm on Reddit data, with various parameters that specify the sketching. In every iteration, we randomly sample 50 users that compute update based on the data available locally, sketch it, and all the updates are averaged. Experiments with sampling $10, 20$, and $100$ clients in each round provided similar conclusions as the following.

In all of the plots, we combine the three components for sketching the updates introduced in Section 3. First, we apply a random rotation to preprocess the local update. Further, 'sketch_fraction' set to either $0.1$ or $1$, denotes fraction of the elements of the update being subsampled.

In the left column, we plot this against the number of iterations of the algorithm. First, we can see that the effect of preprocessing with the random rotation has significantly positive effect, particularly

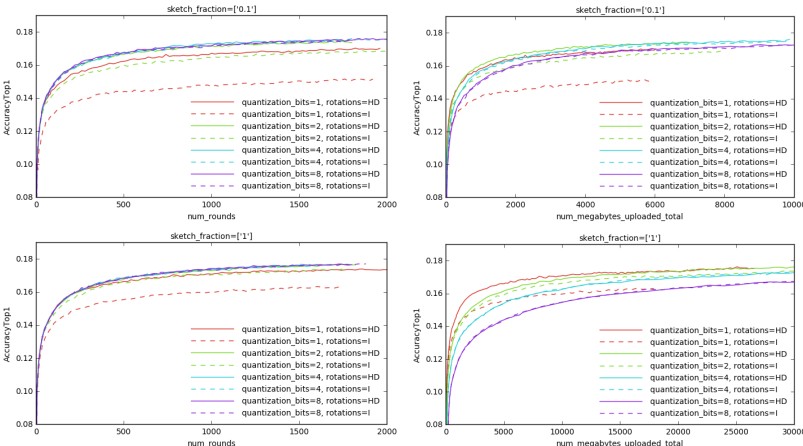

Figure 4: Comparison of sketched updates, training a recurrent model on the Reddit data, randomly sampling 50 clients per round.

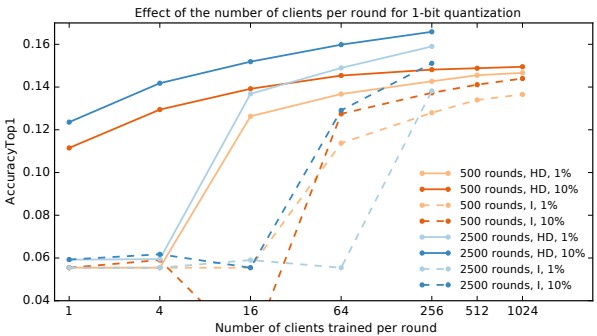

Figure 5: Effect of the number of clients used in training per round.

with small number of quantization bits. It is interesting to see that for all choices of the subsampling ratio, randomized Hadamard transform with quantization into 2 bits does not incur any loss in performance. An important measure to highlight is the number of rounds displayed in the plots is 2000. Since we sample 50 users per round, this experiment would not touch the data of most users even once! This further strengthens the claim that applying Federated Learning in realistic setting is possible without affecting the user experience in any way.

In the right column, we plot the same data against the total number of megabytes that would need to be communicated by clients back to the server. From these plots, it is clear that if one needed to primarily minimize this metric, the techniques we propose are extremely efficient. Of course, neither of these objectives is what we would optimize for in a practical application. Nevertheless, given the current lack of experience with issues inherent in large scale deployment of Federated Learning, we believe that these are useful proxies for what will be relevant in a practical application.

Finally, in Figure 5, we study the effect of number of clients we use in a single round on the convergence. We run the Federated Averaging algorithm for a fixed number of rounds (500 and 2500) with varying number of clients per round, quantize updates to 1 bit, and plot the resulting accuracy. We see that with sufficient number of clients per round, 1024 in this case, we can reduce the fraction of subsampled elements down to 1%, with only minor drop in accuracy compared to 10%. This suggests an important and practical tradeoff in the federated setting: one can select more clients in each round while having each of them communicate less (e.g., more aggressive subsampling), and obtain the same accuracy as using fewer clients, but having each of them communicate more. The former may be preferable when many clients are available, but each has very limited upload bandwidth — which is a setting common in practice.

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
