# OpenReview forum: "Federated Learning: Strategies for Improving Communication Efficiency"
_ICLR.cc/2018/Conference — Reject_

### Official Review · AnonReviewer3 · 2017-11-27
**Reducing uplink communication in distributed setting: a strategy that works well, could use more clarity and insight.**

**Rating:** 5
**Confidence:** 3

**Review:**

This paper proposes several client-server neural network gradient update strategies aimed at reducing uplink usage while maintaining prediction performance.  The main approaches fall into two categories: structured, where low-rank/sparse updates are learned, and sketched, where full updates are either sub-sampled or compressed before being sent to the central server.  Experiments are based on the federated averaging algorithm.  The work is valuable, but has room for improvement.

The paper is mainly an empirical comparison of several approaches, rather than from theoretically motivated algorithms.  This is not a criticism, however, it is difficult to see the reason for including the structured low-rank experiments in the paper (itAs a reader, I found it difficult to understand the actual procedures used.  For example, what is the difference between the random mask update and the subsampling update (why are there no random mask experiments after figure 1, even though they performed very well)?  How is the structured update "learned"?  It would be very helpful to include algorithms.

It seems like a good strategy is to subsample, perform Hadamard rotation, then quantise.    For quantization, it appears that the HD rotation is essential for CIFAR, but less important for the reddit data.  It would be interesting to understand when HD works and why,  and perhaps make the paper more focused on this winning strategy, rather than including the low-rank algo.

If convenient, could the authors comment on a similarly motivated paper under review at iclr 2018:
VARIANCE-BASED GRADIENT COMPRESSION FOR EFFICIENT DISTRIBUTED DEEP LEARNING

pros:

- good use of intuition to guide algorithm choices
- good compression with little loss of accuracy on best strategy
- good problem for FA algorithm / well motivated
-

cons:

- some experiment choices do not appear well motivated / inclusion is not best choice
- explanations of algos / lack of 'algorithms' adds to confusion

a useful reference:

Strom, Nikko. "Scalable distributed dnn training using commodity gpu cloud computing." Sixteenth Annual Conference of the International Speech Communication Association. 2015.

---

> ### Author Response · Authors · 2017-12-15
> **Response**
>
> Thank you for your feedback, helping us see which parts are not communicated clearly enough. Please see also our response to all reviewers above.
>
> Difference between Random Mask and Subsampling - These are techniques presented in Sections 2 and 3, respectively. For Random Mask, we compute and apply the gradient only to pre-selected coordinates. For subsampling, we compute and apply gradients without constraint, and subsample at the end. If we were to run the local optimization for just a single gradient update, these updates/gradients would be identical; however, the subsequent gradients computed locally before communicating, would already be different as they would be computed in different points.
> We did not continue with Random Mask experiments further, as it is not straightforward to use this jointly with the other techniques, such as structured random rotation. The most important gains were obtained as a combination of these multiple techniques. If we trained the Random Mask update in the rotated space, we would make the training procedure significantly more expensive, as applying the rotation would be necessary for every gradient computation. However, applying structured random rotation only once at the end is negligible compared to total cost of training.
> We will make these points more clear in the submission.
>
> CIFAR vs. Reddit data
> We don’t intend to emphasize the CIFAR data too much, as it is relatively small, and artificially partitioned by us to fit the setting of FL. The Reddit dataset comes with natural user partition and is much more reflective of actual application in practice. The HD rotation do actually improve performance significantly - this is perhaps more clearly visible in Figure 5 where we experiment with more clients per round, and can compress very aggressively - 1% subsampling and 1 bit quantization! We will stress the Reddit experiments more in final version.
>
> Pointer to other ICLR submissions (Variance-based Gradient Compression for Efficient Distributed Deep Learning) - Note there is also another submission in similar spirit (Deep Gradient Compression: Reducing the Communication Bandwidth for Distributed Training)
> In both of these works, the central part of proposed techniques keeps track of compression/quantization error incurred in previous rounds, and adds this in the current round before applying compression/quantization. This is not applicable in the setting of Federated Learning, as we cannot remember such errors - think of the billions of eligible phones in the world, but selecting only thousands to participate in a given round.

---

### Official Review · AnonReviewer1 · 2017-11-28
**The paper examines techniques to lower the communication of distributed model updates in a federated setup. The authors focus on low-rank, sparsified, and quantized updates. There are several interesting experiments, but comparisons with state-of-the-art quantization techniques are missing.**

**Rating:** 7
**Confidence:** 5

**Review:**


The authors examine several techniques that lead to low communication updates during distributed training in the context of Federated learning (FL). Under the setup of FL, it is assumed that training takes place over edge-device like compute nodes that have access to subsets of data (potentially of different size), and each node can potentially be of different computational power. Most importantly, in the FL setup, communication is the bottleneck. Eg a global model is to be trained by local updates that occur on mobile phones, and communication cost is high due to slow up-link.

The authors present techniques that are of similar flavor to quantized+sparsified updates. They distinguish theirs approaches into 1) structured updates and 2) sketched updates. For 1) they examine a low-rank version of distributed SGD where instead of communicating full-rank model updates, the updates are factored into two low rank components, and only one of them is optimized at each iteration, while the other can be randomly sampled.
They also examine random masking, eg a sparsification of the updates, that retains a random subset of the entries of the gradient update (eg by zero-ing out a random subset of elements). This latter technique is similar to randomized coordinate descent.

Under the theme of sketched updates, they examine quantized and sparsified updates with the property that in expectation they are identical to the true updates. The authors specifically examine random subsampling (which is the same as random masking, with different weights) and probabilistic quantization, where each element of a gradient update is randomly quantized to b bits.

The major contribution of this paper is their experimental section, where the authors show the effects of training with structured, or sketched updates, in terms of reduced communication cost, and the effect on the training accuracy. They present experiments on several data sets, and observe that among all the techniques, random quantization can have a significant reduction of up to 32x in communication with minimal loss in accuracy.

My main concern about this paper is that although the presented techniques work well in practice, some of the algorithms tested are similar algorithms that have already been proven to work well in practice. For example, it is unclear how the performance of the presented quantization algorithms compares to say  QSGD [1] and Terngrad [2]. Although the authors cite QSGD, they do not directly compare against it in experiments.

As a matter of fact, one of the issues of the presented quantized techniques (the fact that random rotations might be needed when the dynamic range of elements is large, or when the updates are nearly sparse) is easily resolved by algorithms like QSGD and Terngrad that respect (and promote) sparsity in the updates.

A more minor comment is that it is unclear that averaging is the right way to combine locally trained models for nonconvex problems. Recently, it has been shown that averaging can be suboptimal for nonconvex problems, eg a better averaging scheme can be used in place [3]. However, I would not worry too much about that issue, as the same techniques presented in this paper apply to any weighted linear averaging algorithm.

Another minor comment: The legends in the figures are tiny, and really hard to read.

Overall this paper examines interesting structured and randomized low communication updates for distributed FL, but lacks some important experimental comparisons.


[1] QSGD: Communication-Optimal Stochastic Gradient Descent, with Applications to Training Neural Networks https://arxiv.org/abs/1610.02132
[2] TernGrad: Ternary Gradients to Reduce Communication in Distributed Deep Learning
https://arxiv.org/abs/1705.07878
[3] Parallel SGD: When does averaging help?
https://arxiv.org/abs/1606.07365

---

> ### Author Response · Authors · 2017-12-15
> **Response**
>
> Thank you for your encouraging review. Below are remarks and responses to your highlighted concerns.
>
> You remark that we achieve up to 32x reduction in communication. We would like to stress that we can achieve a lot more - with combining subsampling, rotations, and quantization without impacting convergence speed. See the extreme in Figure 5 where we subsample 1% of the elements and then quantize to 1bit (3200x on the compressed layers, although with a drop in performance).
>
> [See also response to all reviewers above for comparison with other methods]
> We also experimented with various adaptive methods which overall provided slightly worse results, before we were aware of the mentioned works. Nevertheless, our very recent preliminary experiment suggests that performance of QSGD improves when we use it with the subsampling and structured random rotation proposed in our work, and is roughly on par with the experiments we present.
>
> Sparsity: Note that if the updates are sparse, it is possible to use a sparse representation first, and then apply the presented techniques to compress list of nonzero values of the sparse representation. It is not ideal, but QSGD degrades in a similar way, as the gaps between non-zero values encoded using Elias coding are no longer necessarily small numbers, making the whole compression slightly weaker.
>
> We agree that it should be possible to do better than averaging within the Federated Averaging of McMahan et al. However, this problem is clearly out of scope for this work, and probably worth a separate paper altogether.

---

### Official Review · AnonReviewer2 · 2017-12-01
**The studied problem seems to be interesting, but there exist several major issues in the paper.**

**Rating:** 5
**Confidence:** 5

**Review:**

This paper proposes a new learning method, called federated learning, to train a centralized model while training data remains distributed over a large number of clients each with unreliable and relatively slow network connections. Experiments on both convolutional and recurrent networks are used for evaluation.

The studied problem in this paper seems to be interesting, and with potential application in real settings like mobile phone-based learning. Furthermore, the paper is easy to read with good organization.

However, there exist several major issues which are listed as follows:

Firstly, in federated learning, each client independently computes an update to the current model based on its local data, and then communicates this update to a central server where the client-side updates are aggregated to compute a new global model. This learning procedure is heuristic, and there is no theoretical guarantee about the correctness (convergence) of this learning procedure. The authors do not provide any analysis about what can be learned from this learning procedure.

Secondly, both structured update and sketched update methods adopted by this paper are some standard techniques which have been widely used in existing works. Hence, the novelty of this paper is limited.

Thirdly, experiments on larger datasets, such as ImageNet, will improve the convincingness.

---

> ### Author Response · Authors · 2017-12-15
> **Reponse**
>
> Thank you for your review, highlighting good motivation and organization of the work. Let us address the three specific issues you highlighted.
>
> The remark on our proposed procedure being heuristic being an issue is in our opinion misplaced.
> The learning procedure (Federated Averaging) is in the first place not the contribution of our paper - it was proposed in McMahan et al., it was shown to work well for large-scale problems and has been successfully deployed in production environment by Google (see McMahan and Ramage), and we build on top of it. This is in line with other optimization techniques for deep learning - they usually have very well understood parallels in the convex setting, but are not really understood in the landscape of deep learning - only empirically observed to typically still work. This procedure is also an extension of existing techniques which are properly analysed in the convex setting - see Ma et al. and Reddi et al. The central part of our contribution does have a proper theoretical justification - see Suresh et al.
>
> While individual building blocks have been used in various works, we are not aware of some of them being used in the context of reducing update size in deep learning. Please see also response to all reviewers above for why some of them are not practical in the standard data-center training.
>
> We have tested our method on the large-scale Reddit dataset, which is highly representative of the types of problems suited to federated learning (unlike ImageNet). The CIFAR experiment can be seen as proof-of-concept but we had to artificially split the dataset into “clients”, and hence does not reflect the practical setting well. The same would be true for ImageNet. The Reddit dataset comes with natural user-based partitioning, and in terms of number of datapoints, is actually much larger than ImageNet.

---

### Author Response · Authors · 2017-12-15
**Response to all reviewers**

We would like to thank all reviewers for their feedback. The following is response relevant to all reviewers, and explains a particular point we will stress more clearly in the submission.

In the best technique we used (subsampling + rotation + quantization), the related recently proposed methods such as QSGD or TernGrad are an alternative to the quantization part, not for the whole procedure. If used separately, they yield a significantly weaker result. Note that the results in QSGD paper, authors generally use more than 1bit per element on average. (see also Corollary 3.3 which promises ~2.8 bits per element asymptotically). In contrast we reduced the communication significantly below 1bit per element.

Our technique yields sparse objects whose sparsity pattern is independent of the objects we are trying to compress. This lets us to communicate only the quantized values, and not the indices those values correspond to - those can be recovered from a shared random seed. Further, applying structured random rotation improves the performance of quantization. These are however more computationally expensive operations (especially rotation), which makes it impractical in the setting of the above mentioned methods (MPI-based GPU-to-GPU communication on small minibatches). Nevertheless, this is a component that significantly improves our performance, and it could actually now become practical also in data-center training, together with the trend shifting to large-batch training (see for instance works on training ImageNet in 1hour, 24 min, 15 mins...)

---

### Decision · Program_Chairs · 2018-01-29
**ICLR 2018 Conference Acceptance Decision**

**Decision:**

Reject

**Comment:**

The authors study the problem of reducing uplink communication costs in training a ML model where the training data is distributed over many clients.   The reviewers consider the problem interesting, but have concerns about the extent of the novelty of the approach.  As the reviewers and authors agree that the paper is an empirical study, and the authors agree that the novelty is in the problem studied and the combination of approaches used, a more thorough experimental analysis would
benefit the paper.